# Promoting language and literacy through shared book reading in the NICU: A scoping review

**Lama K. Farran**[1]*, **Sharon L. Leslie**[2], **Susan N. Brasher**[3]

**1** Department of Counseling, Higher Education, & Speech-Language Pathology, University of West Georgia, Carrollton, Georgia, United States of America, **2** Woodruff Health Sciences Center Library, Emory University, Atlanta, Georgia, United States of America, **3** Nell Hodgson Woodruff School of Nursing, Emory University, Atlanta, Georgia, United States of America

* lfarran@westga.edu

## Abstract

### Background

Infants in the neonatal intensive care unit (NICU) are at a heightened risk for language and literacy delays and disorders. Despite the well-established empirical support for early shared reading, the available evidence to date has been scant, revealing mixed results. This study sought to characterize current research on shared reading in the NICU using a scoping review methodology.

### Methods

Studies were eligible for inclusion if they were peer-reviewed, written in the English language, focused on human infants in the NICU, and published between January 1, 2003, and December 31, 2023. No population age range was applied, and quantitative, qualitative, or mixed methods designs were considered. Database searches yielded 338 articles with only eight articles meeting eligibility criteria for inclusion.

### Conclusion

In spite of a modest number of studies on this topic, utilizing limited methodologies, the evidence from this scoping review shows the benefits of shared reading for infants and their caregivers during their NICU stay. Expanding such efforts by embedding shared reading as part of standard practice is recommended.

## Introduction

The ability to read is uniquely human and built on a foundation of oral language [1]. It begins very early in life and emerges from the back-and-forth communicative exchanges between infants and their caregivers (e.g., mother, father), especially in the context of shared book reading [2,3]. Mounting evidence suggests that the cause of reading disorders is multifactorial [4], with complex interactions among biological, ecological, and developmental

**Data availability statement:** All relevant data are within the paper and its Supporting Information files.

**Funding:** This research has been funded by a grant from the Deal Center for Early Language and Literacy to Lama K. Farran.

influences playing a paramount role [5]. Therefore, attempts at early identification and intervention must take into account the multitude of factors–including protective and risk factors–that are paramount for a preventive model of reading and its disorders [6]. This is especially the case for infants in the Neonatal Intensive Care Unit (NICU), who come to the world with a host of risk factors (e.g., genetic, medical, environmental) that potentially stunt their development of language, negatively impacting their opportunities to develop adequate reading. Interventions such as shared reading in the NICU aimed at offsetting the risk for reading delays and disorders are thus considered protective and necessary. They constitute the focus of this article.

Currently, 1 in every 10 infants is born preterm (i.e., before the completion of 37-week gestation) [7]. While the number of preterm births in the United States decreased by 1% from 2021 to 2022, racial and ethnic differences in preterm births did not, particularly for Black women (14.6%), which is estimated to be 50% higher compared to White (9.4%) and Hispanic (10.1%) women, representing significant racial and ethnic disparities in preterm birth rates [8]. Infants who are born prematurely typically stay in the NICU, along with other infants who require intensive medical care (e.g., low birth weight, very low birth weight, and medically complex). These infants are at a heightened risk for delays and disorders across multiple developmental domains brought about by their medical condition, less-than-optimal NICU environment, and concomitant morbidity. Therefore, the language and reading input that infants receive from their caregivers and healthcare team (e.g., registered nurses, speech-language pathologists, physical therapists, occupational therapists) is foundational for supporting caregivers of NICU infants [9] and equipping these infants with social-emotional and cognitive skills that provide them with an advantage in their development toward adulthood [10–12], thereby offsetting the negative sequelae of their NICU stay. Through shared book reading, caregivers intuitively infuse language during their interactions with their infants, potentially improving children's developmental outcomes in reading, cognition, and language in early childhood [13–21]. Similarly, the healthcare team in the NICU plays a foundational role in bolstering early childhood development through strategies aimed at supporting infant-mother attachment, as well as mothers' emotions, empowerment, and participation in the care process [12].

The extant research indicates that shared book reading in the first year may be associated with future health and quality of life of caregivers and their infants across developmental domains [22,23], namely mental, emotional health, language, and literacy. Notably, shared book reading in the NICU has been shown to promote mother-infant bonding, decrease postnatal maternal stress, and encourage continued reading throughout childhood [8]. Despite the reported benefits of shared book reading interventions between parents and their infants in the NICU, the evidence to date has been limited, focusing on interventions that are brief in duration [24]. Thus, there is a need for changing the NICU environment to promote interventions that are effective and long lasting. One such change would be to embed shared book reading within routine developmental care in the NICU as standard practice. This would translate into capacity building from staff to caregivers during the NICU stay, potentially contributing to positive effects on language and literacy and increasing the sustainability of overall whole child outcomes.

This scoping review focuses on shared reading and examines interventions aimed at improving the outcomes of infants in the NICU. We examine the following research question: What is known from research about shared reading interventions in the NICU? First, we discuss key variables that directly impact shared reading, including the NICU environment and caregiver-infant interaction. We then review interventions and programs that address shared reading in the NICU directly, and end with strengths and limitations of the current research

on this topic, as well as recommendations for future emphasis on translational research and implementation science.

## NICU Environment

Infants born preterm, low birth weight (< 2500 grams), very low birth (<1500 grams), acutely ill (e.g., sepsis, respiratory distress syndrome), and with congenital anomalies constitute the majority of neonatal admissions to the NICU [25]. While intended to be supportive of fragile infants, the NICU environment can also be described as a stressful environment filled with high- and mid-frequency sounds (e.g., monitors beeping, ventilator sounds, voices of staff and other families) [26] and isolating conditions for the infant (e.g., incubators) and families (e.g., feelings of alienation) [26]. Hospitalization in the NICU contributes to limited and impaired interaction between parents and their preterm infants due to stress and tension [27]. As a result, infants in the NICU receive low levels of language exposure and high levels of sound exposure, which have been linked with poorer growth, cognition, language and motor outcomes [28]. Conversely, research has found that increasing language exposure in the NICU is associated with better language outcomes in infants [29] and that the NICU can serve as a protective factor when focusing on responsive parenting and rich language and literacy input early in life [30].

## Caregiver-infant interaction

Another factor that impacts infants and caregivers is caregiver-infant interactions, which is paramount to infants' development of social and lexical skills [31,32]. Specifically, live maternal speech is thought to be associated with promoting infants' autonomic stability, including respiration and heart rate [33], further bolstering the importance of mother-child interaction and supporting the health of the mother-infant dyad. By contrast, maternal distress associated with postpartum depression negatively impacts mother-child interactions, thus putting infants at a greater risk for developmental delays [34].

## Shared Book Reading

Perhaps one of the best contexts that promotes caregiver-infant interactions in the NICU is shared book reading, connecting caregivers and their infants and providing them a way to participate in the caregiving experience [35]. Given the importance and benefit of early language and reading input, shared reading is thought to play an instrumental role in closing developmental gaps for infants in the NICU and consists of caregivers talking with the infant, actively engaging the infant in the reading activity through touch, use of motherese (also known as baby talk), singing, rhyming, and visual stimulation. These practices, when implemented by either caregivers or NICU staff members, have been found to benefit the language and literacy development of preterm infants, while strengthening parent-child relationships [35,36]. As they get older, children who engage in shared book reading with caregivers will utilize increasingly more sophisticated language as evidenced by a higher frequency of questions and the presence of a larger vocabulary base [37].

## Theoretical Framework

This scoping review is cast in two developmental approaches: (a) the usage-based theory [38–40] and (b) the eco-biodevelopmental model of emergent literacy [5] (Fig 1). The usage-based theory argues that language is an emergent phenomenon that relies on patterns of use and communicative exchanges. Accordingly, reading is an outgrowth of language experience and language use, with daily, frequent communicative bids serving as its foundation. As

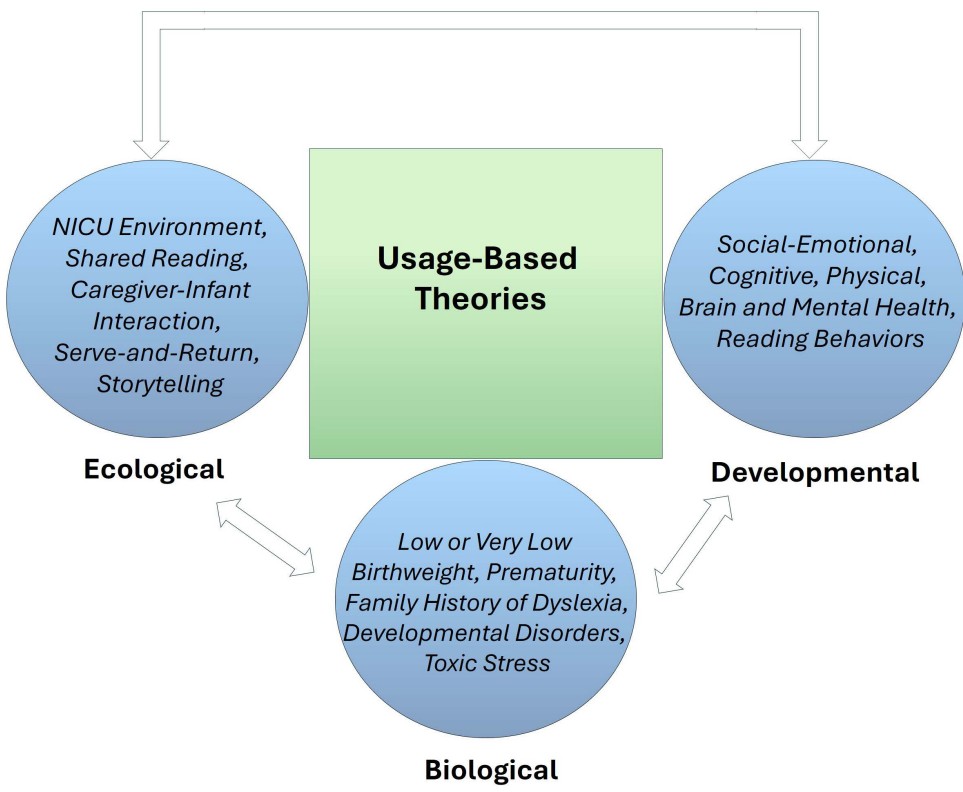

**Fig 1.** Eco-biodevelopmental usage-based model of emergent literacy.

such, the more caregivers and infants engage in shared reading practices in the NICU, the more connected (e.g., social-emotional, skin-to-skin) they become as evidenced by improved turn-taking skills, and increased willingness to read and connect in the future. Improved language use (pragmatic language) thus gives rise to better language structure (syntax), which translates to other components of language such as meaning (semantics), speech sounds (phonology), and word form (morphology) across spoken (oral language) and written (reading and writing) modalities. These interactive patterns of language use tend to offset the negative sequelae of the NICU environment including NICU ambient noise, maternal depression, and infant distress.

The eco-biodevelopmental model of emergent literacy which also informs this review, posits that reading has its origins in infancy and the early years that precede formal schooling, and that reading outcomes depend on the orchestration between ecological, biological, and developmental factors. As such, ecological contributions include home literacy practices, shared reading, caregiver-infant interaction, talking with infants, storytelling, and play. When applied to the NICU environment, the level and intensity of ambient noise, socioeconomic status (SES) of the family (e.g., high or low, poverty), maternal education, and concomitant factors such as epigenetic mechanisms associated with toxic stress, likely impact the degree to which caregivers engage in language and shared reading practices with their infants. While research suggests that caregivers from low SES backgrounds tend to have children with a higher risk for reading difficulties, read less frequently, and initiate reading with their children at older ages [41,42], further research is needed to explore the role of protective factors in shaping the developmental outcomes of these children.

In addition to the ecological influences, infants in the NICU present with a host of biological factors including a familial risk for reading disorders, predisposing them to reading difficulties. Moreover, comorbid conditions such as hearing loss, prematurity, low or very low birth weight, and a family history of autism spectrum disorders (ASD) (e.g., having siblings who are diagnosed with ASD; family with a broader autism phenotype) or dyslexia serve as another layer of difficulties for infants in the NICU. Notably, bidirectional relationships exist between ecological and biological factors and influence infants' (and mothers') social-emotional development and physical, brain, and mental health. The more caregiver support (e.g., social-emotional, maternal vocal responsivity, language input, shared reading) these infants receive early in their development, the better their language and reading outcomes. Thus, caregiver-infant dyads who engage in shared reading practices that are rich in quantity and quality in the NICU will likely have better experience with reading, resulting in more frequent reading practices. Such an early exposure to print and language leads to reading readiness, positively impacting infants' brain development [43–45] as well as their physical and mental health.

## Materials and methods

### Study design

This review was organized and reported in accordance with the Preferred Reporting Items for Systematic Reviews and Meta-Analyses Extension for Scoping Reviews (PRISMA-ScR) [46] [S1 Checklist]. A scoping review was chosen to allow for inclusion of studies from any methodology type and to report broadly on the literature in the field. No protocol was registered.

### Inclusion/exclusion criteria

This study sought to identify gaps in the literature and gather research evidence that characterizes shared reading in NICUs. Eligibility criteria were set to include studies that were peer-reviewed, available in the English language, involved infants in the NICU, included an intervention of reading books aloud or narrating a physical or e-book, and were published between January 1, 2003, and December 31, 2023. No population age range limit was applied. The study considered all studies that had used quantitative, qualitative, or mixed methods designs.

### Database search strategy

An experienced medical librarian (SL) developed a comprehensive literature search strategy with input from the research team to identify articles meeting the study criteria. Pre-identified sentinel articles were hand searched for keywords relating to the study objectives. An initial search strategy was drafted for testing in PubMed. Those results were assessed, and additional terms were selected from the titles, abstracts, and MeSH terms. A second draft search was created with the added terms and again tested in PubMed.

The searches combined controlled vocabulary supplemented with keywords related to the concepts of neonatal intensive care units (e.g., NICU, premature) and shared reading (e.g., reading aloud, language exposure). Searches were initially undertaken December 8, 2023. Full search strategies for each database may be found in S1 Supporting Information. The search strategy for PubMed may be found in Fig 2.

Nine bibliographic databases were searched: APA PsycINFO (EBSCOhost), Child Development and Adolescent Studies (EBSCOhost), Cumulative Index to Nursing & Allied Health Literature (CINAHL, EBSCOhost), Embase.com (Elsevier), ERIC (EBSCOhost), Health

("Neonatal intensive care unit*"[tw] OR "neo-natal intensive care unit"[tw] OR "neonatal ICU*"[tw] OR NICU*[tw] OR "Intensive Care Units, Neonatal"[Mesh] OR Premature[tw] OR prematurity[tw] OR Preterm[tw] OR newborn*[tw] OR "Infant, Newborn"[Mesh]) **AND** ("shared reading"[tw] OR "shared book reading"[tw] OR "Reach Out and Read"[tw] OR "reading aloud"[tw] OR "read aloud"[tw] OR "reading session*"[tw] OR "parental reading"[tw] OR "language exposure"[tw] OR "speech exposure"[tw] OR ("Reading"[Mesh] AND "Parent-Child Relations"[Mesh])) **AND** ("2003/01/01"[PDAT] : "2023/12/31"[PDAT]) **AND** english[Filter]

**Fig 2.** PubMed search strategy.

Source: Nursing/Academic (EBSCOhost), Linguistics and Language Behavior Abstracts (Pro-Quest), PubMed, and Scopus (Elsevier). After the searches, all identified records were collated and uploaded directly into Covidence, a systematic review software.

## Results

### Study selection

A total of 338 articles were identified through the database searches and uploaded to Covidence. Covidence identified 195 duplicates, leaving 143 records for title/abstract screening. The titles and abstracts were reviewed by two independent reviewers (LKF, SB) according to the inclusion/exclusion criteria. Disagreements between reviewers were resolved by discussion and consensus agreement. Of these records, 129 were excluded for irrelevancy, leaving 14 eligible for full-text review. During the full-text review, the reviewers independently evaluated each article and excluded six articles, leaving eight articles that met all the eligibility criteria for inclusion in this study. The review and selection processes for the studies are summarized in the diagram in Fig 3.

**Findings related to shared reading in the NICU.** The eight articles that met inclusion criteria are listed in Table 1. Despite our search spanning 20 years of research, a majority of the literature on shared reading in the NICU was published in the previous year (2023). While the importance of shared reading in the NICU is well documented [47,48], the lag in evaluation and dissemination in peer-reviewed journals represents an area of need for improvement. All eight articles emphasized shared reading in the NICU; however, the outcomes as well as the methodologies that were used to measure those outcomes varied. Most of the studies leveraged a survey study design alone [49–52] or in combination with a mixed methods design [23,53]. Two studies utilized objective and observable measures, such as General Motor Assessment of Movement and Bayley Scales of Infant Development [54], as well as heart rate, respiratory rate, and oxygen saturation [55]. All studies provided books to the families including general children's books [55] and through formalized programs such as Reach Out and Read [52,54], R.E.A.D (Read to, Enjoy, And Develop) Your Baby program [49], the Little Reader's Read-a-thon [50], NICU Bookworm program [51], the Books for Babies reading program [23], and Read-a-Latte read-a-thon with books from Cleveland Kids' Book Bank, Reach Out and Read, the Literacy Cooperative at Dolly Parton's Imagination Library, and Project NICU [53].

The study data was extracted by two reviewers (LKF, SB). Studies spanned several countries and regions, including Brooklyn, New York (U.S.) [49], New South Wales (Australia) [50], Cincinnati, Ohio (U.S.) [51], Belagavi (India) [54], Montreal (Canada) [23], Cleveland, Ohio (U.S.) [53], Boston, Massachusetts (U.S.) [52], and Washington D.C. (U.S.) [55]. Two studies emphasized use of books in a language other than English including Levesque et al. [52] using

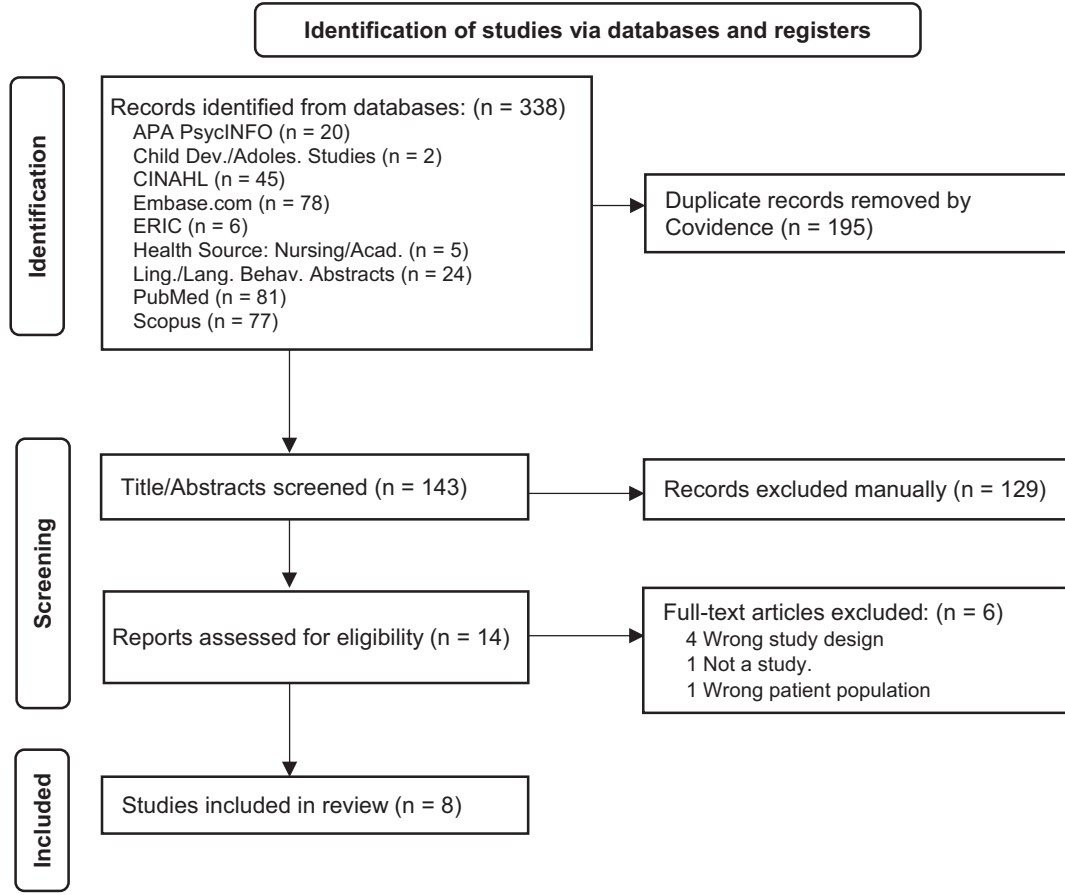

**Identification of studies via databases and registers**

**Identification**

Records identified from databases: (n = 338)
APA PsycINFO (n = 20)
Child Dev./Adoles. Studies (n = 2)
CINAHL (n = 45)
Embase.com (n = 78)
ERIC (n = 6)
Health Source: Nursing/Acad. (n = 5)
Ling./Lang. Behav. Abstracts (n = 24)
PubMed (n = 81)
Scopus (n = 77)

Duplicate records removed by Covidence (n = 195)

**Screening**

Title/Abstracts screened (n = 143)

Records excluded manually (n = 129)

Reports assessed for eligibility (n = 14)

Full-text articles excluded: (n = 6)
4 Wrong study design
1 Not a study.
1 Wrong patient population

**Included**

Studies included in review (n = 8)

**Fig 3.** PRISMA flow diagram illustrating the study selection.

books in the family's primary language, and Kale and Deshpande [54] using books in Hindi, Marathi, and Kannada, in addition to English.

Studies also varied widely in terms of their use of tracking methods to document parent reading practices. Levesque et al. [52] used calendars and stickers where parents were asked to record how often they read to their infants. They also used electronic medical records (EMR) to track parental presence in the NICU. Lariviere and Rennick [23] assessed parent reading to their infants using the Parent-Infant Activity Sheet, a 20-item questionnaire that assesses the frequency and duration of infant care activities, including parent-infant reading. Kale and Deshpande [54] tracked parental adherence to regular intervention at home using weekly phone calls and during the monthly visits to the high-risk baby clinic. Such variability would be important to document as it tends to reduce the replicability of the studies, rendering measurement of tracking parental reading practices less accurate.

Moreover, the studies differed with respect to the medical or caregiving team participating in the shared reading program. For example, studies ranged from teams of professionals (neonatologist, NICU nurse manager, ROR coordinator, and data collector) [52], to a single professional like a nurse [23] or a pediatric physiotherapist [54]. These differences in who is involved in the caregiving in the NICU introduces a difficult-to-control variable which potentially interferes with comparability between the reviewed studies.

**Table 1. Summary of included studies.**

| Citation | Setting and Population | Study Design | Intervention | Results |
|---|---|---|---|---|
| [49] | Caretakers of infants in the NICU. The pre- survey consisted of 50 families (2019). Post-survey data collection consisted of 12 families due to COVID-19. | Pre- and Post- Survey. | The R.E.A.D. (Read to, Enjoy, And Develop) Your Baby program. The campaign consists of giving baby books to families with an infant in the NICU >7 days and bi-weekly for chronically hospitalized infants. | Shared reading was occurring for only a minority of infants in the NICU before the R.E.A.D. Your Baby campaign. Significant changes were observed in survey responses centered around language use (talking, singing, or reading) by parents in the NICU from 25.2% in the pre-survey to 41.7% in the post-survey (p = .024). Shared reading in the NICU increased from 22% of parents in the pre-survey to 91.7% in the post-survey (p = .00000). |
| [50] | Parents of infants (n = 35) in a surgical NICU. | Cross-sectional non-identical sampling survey pre- and post-intervention. | Read-a-thon program. | There was an increase in reading recommended post-intervention (86%) compared to pre-intervention (76%). Reading in the first week of life significantly increased from 43% pre-intervention to 90% of post-intervention (p = 0.022). Encouraging factors for reading included privacy, having books available and being able to hold their baby. Discouraging factors included other people hearing them read, if their baby is sleeping, and if there were too many people near their baby. The discouraging factor of other people hearing them read decreased significantly from 35% pre-intervention to 8% post-intervention (p = 0.042). |
| [51] | 317 infants: unexposed comparison group (n = 187); intervention group (n = 130). | Parent survey. | Bookworms book-sharing reading intervention in the NICU. | Parents in the intervention group reported reading aloud > or equal to 3-4 days per week more to their infants in the NICU compared to parents in the unexposed comparison group (34.5% vs 51.5%; p = .002; aOR, 2.2; 95% CI, 1.2–4.0). |
| [54] | 18 infants, 7 or > days in NICU, Apgar score of more than 6 at first and fifth minute. | Pre- and Post- experimental study design. | Evaluated effectiveness of Reach Out and Read (ROR) on communication and generalized movement at a tertiary hospital in India over a period of 6 months. | Non-parametric statistics (Wilcoxon matched pair test) showed improvement in general movement over time (p = .0277; p = .0431). However, no significant increase in expressive or receptive communication outcomes (p > .05). |
| [23] | 116 infants recruited in the NICU: Intervention group (n = 59); historical control group (n = 57). | Mixed methods design using a non-randomized, participant blinded intervention study using a historical control group. Pre- and Post- study design with survey and qualitative interviews. | Parental reading intervention in the NICU. Parents selected a book and read every day as they held their infant at bedside or in the incubator. | Increased frequent reading (majority reported reading more than three times per week) in the intervention group 55.9% compared to the control group 17.5% (p < .001); evident three months after discharge. Parents reported a higher sense of control. |
| [53] | 1,255 infants admitted to a level IV NICU and step-down NICU. | Qualitative (interviews, open-ended surveys, document reviews). | Read-a-Thon (Read-a-Latte) spanning 10 days. Books were distributed to families of all infants in the NICU. Healthcare professionals and caregivers were encouraged to read to infants. | A total of 663 reading sessions were logged during the 10-day Read-a-Thon. Six qualitative themes were identified: Motivation, emotional response to the program, benefits and outcomes, barriers, facilitators, and future of literacy promotion in the NICU. |
| [52] | 98 preterm infants born between March 1 and December 31, 2015, < 37-week gestation. | Descriptive statistics (percent enrolled, read to, etc.) Anonymous parent survey. | Pilot study of Reach Out and Read (ROR). Parental reading with 95% of infants read to using books in the mothers' primary language (English, Spanish, Portuguese, French, Haitian Creole, Vietnamese). | The median percentage of families reading to their baby increased from 0% before and 59% after the ROR program. |
| [55] | 18 Preterm infants (23–31-week gestation). | Prospective unblinded pre- and post- study design. | Parental (mothers and fathers) reading intervention in the NICU. Comparison between live and recorded reading. | Fewer desaturation events (reduced Oxygen) of less than 85% during parental reading than prior to reading exposure (p = .0001). These effects persisted up to 1 hour after reading exposure. |

## DISCUSSION

Our scoping review revealed that infants benefit from shared reading during their NICU stay. Generally, the shared reading programs place a heavy emphasis on shared book reading in the NICU, educating parents on the benefits of shared reading, how to effectively read to their child, and how to implement language from books into daily routines, while facilitating caregiver access by placing children's books throughout the NICU environment. Beyond the NICU, the Reach Out and Read program [52] provides participants with two free books initially, followed by one book per month. While this program is reported to promote shared book reading by equipping parents with the resources, it does not include a method of tracking shared reading habits beyond the NICU, nor does it confirm that the caregivers integrated shared book reading into their daily life.

The available research suggests that shared reading program implementations to date have largely emphasized caregiver training in the NICU, with the rationale that caregivers would be more likely to continue shared book reading practices that positively impact infants' language and reading outcomes after being discharged [23]. However, given the lack of mechanisms in place for motivating the parents and holding them accountable for implementing shared reading, the long-term benefits of shared reading *after* NICU discharge could not be substantiated, rendering the effectiveness of caregiver training tenuous. Furthermore, the dearth of longitudinal investigations and the limitations inherent in the methodologies employed in the studies on shared reading in the NICU provide additional support for the lack of effective shared reading interventions at this time.

This review indicates that shared reading programs and initiatives are available, yet lack standardization, serving as add-ons to the NICU stay rather than an integral part embedded into the existing routine practices of the NICU. As such, shared reading programs are introduced, incorporated into hospital NICUs, and then replaced by other shared reading programs, resulting in temporary, unsustainable changes in the NICU. While the practices that many shared reading programs promote are beneficial, caregivers and infants would benefit most from having shared book reading woven into their NICU stay as an essential routine element, that is reflective of their racial, ethnic, and cultural background.

When considering the extant research, the review findings align with previous studies which consider the need for a preventive approach based on current research evidence. Recent findings suggest structural brain (cerebellar) anomalies [56], lack of processing efficiency, and reduced difficulties, including cognition and reading in children who were in the NICU [57]. Efforts must be directed to identify these children as early as possible to mitigate the cascading learning sequelae that derail their developmental trajectories, negatively impacting their future cognitive outcomes, including reading [56,57]. Furthermore, enhancing parent language input not only boosts infants' volubility (i.e., number of vocalizations) in the NICU [29] but also bolsters their speech and language outcomes as evidenced by increased infant volubility at 30 weeks and their conversational turns with their parents at 32 weeks compared to language input from other adults [29], potentially maximizing infants' developmental outcomes beyond the NICU stay.

One primary aim of this review was to evaluate the existing literature and programs that target shared book reading in the NICU. By uncovering the strengths and limitations that exist in the literature, we can expand research efforts to inform practice on shared reading. Future studies must move beyond creating or expanding a program that educates parents on shared book reading; instead, they should aim to train NICU staff (e.g., physicians, nurses, clerks, administrators) using translational research and implementation science principles to embed shared book reading as standard practice within the NICU. We believe that such

training will likely address the issue of language and reading delays more systematically, with interventions beginning as early as possible. A "train the trainer" program targeting not only parents, but all individuals who interact with the infants in the NICU, will help ensure shared reading practices are encouraged and enforced permanently, beyond the entry and exit of NICU stay. This would maximize capacity transfer from staff to caregivers and ultimately result in improved developmental language and reading outcomes for all infants.

### Strengths and Limitations

While this scoping review had many strengths, it also had some limitations. In terms of strengths, having the expertise of a medical librarian added authority to the development of our search strategies and database choices. Furthermore, involving the librarian who is representative of end users and consumers of this research was important as stakeholder engagement was central to our review. We believe such an addition serves to ensure relevance, contextualization, and uptake of the research findings [58]. The use of an electronic tool, Covidence, helped increase the rigor of and decrease bias in our review.

With respect to limitations, they centered on four main areas. First, the review identified only eight studies that met inclusionary criteria, pointing to the dearth of research on this topic. Second, while our search targeted shared reading in the NICU, most of the studies focused on premature infants at the expense of other medically fragile infants who would be optimal candidates for shared reading interventions. Third, the methodological rigor was tenuous as the studies reviewed employed only pre- and post-study designs, descriptive statistics, or exploratory qualitative analyses. Fourth, the construct of shared reading was confined to shared book reading, precluding the inclusion of families from culturally and linguistically diverse backgrounds who are likely engaging in shared reading in other ways (e.g., using digital stories, oral storytelling that are more representative of their cultural heritage). Furthermore, sources in languages other than the English language were excluded due to the authors' language limitations and lack of funds for translation. Such findings provide opportunities for future investigations that address the needs of all infants and parents in the NICU.

### CONCLUSION

In conclusion, shared reading in the NICU has been found to be beneficial for infants and their caregivers during their NICU stay. The cumulative evidence based on this scoping review supports reading to infants in the NICU and calls for an expansion of such efforts by embedding shared reading as part of standard practice, involving both caregivers and healthcare providers. An understanding of the neurobiology of infants in the NICU and shared reading itself as a developmental process can help us design systemic, sustainable interventions that consider key variables, including diverse eco-biological forces that contribute to and are shaped by the dynamic nature of the developing infant.

### Supporting Information

**S1 Table. Summary of Included Studies.**
(DOCX)

**S1 Supporting Information. Full Search Strategies.**
(DOCX)

**S1 Checklist. Preferred Reporting Items for Systematic reviews and Meta-Analyses extension for Scoping Reviews (PRISMA-ScR) checklist.**
(DOCX)

## Author contributions

**Conceptualization:** Lama K. Farran.

**Data curation:** Lama K. Farran.

**Formal analysis:** Sharon L. Leslie.

**Funding acquisition:** Lama K. Farran.

**Investigation:** Lama K. Farran.

**Methodology:** Lama K. Farran, Sharon L. Leslie, Susan N. Brasher.

**Project administration:** Lama K. Farran.

**Resources:** Lama K. Farran, Sharon L. Leslie, Susan N. Brasher.

**Writing – original draft:** Lama K. Farran, Susan N. Brasher.

**Writing – review & editing:** Lama K. Farran, Sharon L. Leslie, Susan N. Brasher.

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
