## [Decision Letter · Decision Letter 0]

22 Oct 2024

PONE-D-23-44086Shared Book Reading in the NICU: A Scoping ReviewPLOS ONE

Dear Dr. Farran

Thank you for submitting your manuscript to PLOS ONE. After careful consideration, we feel that it has merit but does not fully meet PLOS ONE’s publication criteria as it currently stands. Therefore, we invite you to submit a revised version of the manuscript that addresses the points raised during the review process.

 Please submit your revised manuscript by Dec 06 2024 11:59PM. If you will need more time than this to complete your revisions, please reply to this message or contact the journal office at plosone@plos.org . Please include the following items when submitting your revised manuscript:

We look forward to receiving your revised manuscript.

Kind regards,

Sanjoy Kumer Dey, M.D

Academic Editor

PLOS ONE

Journal Requirements: When submitting your revision, we need you to address these additional requirements. 1. Please ensure that your manuscript meets PLOS ONE's style requirements, including those for file naming. The PLOS ONE style templates can be found at https://journals.plos.org/plosone/s/file?id=wjVg/PLOSOne_formatting_sample_main_body.pdf and https://journals.plos.org/plosone/s/file?id=ba62/PLOSOne_formatting_sample_title_authors_affiliations.pdf 2. Thank you for stating the following financial disclosure: "This research has been funded by a grant from the Deal Center for Early Language and Literacy to Lama K. Farran." Please state what role the funders took in the study.  If the funders had no role, please state: ""The funders had no role in study design, data collection and analysis, decision to publish, or preparation of the manuscript."" If this statement is not correct you must amend it as needed. Please include this amended Role of Funder statement in your cover letter; we will change the online submission form on your behalf. 3. We note that your Data Availability Statement is currently as follows: All relevant data are within the manuscript and its Supporting Information files. Please confirm at this time whether or not your submission contains all raw data required to replicate the results of your study. Authors must share the “minimal data set” for their submission. PLOS defines the minimal data set to consist of the data required to replicate all study findings reported in the article, as well as related metadata and methods (https://journals.plos.org/plosone/s/data-availability#loc-minimal-data-set-definition). For example, authors should submit the following data: - The values behind the means, standard deviations and other measures reported;- The values used to build graphs;- The points extracted from images for analysis. Authors do not need to submit their entire data set if only a portion of the data was used in the reported study. If your submission does not contain these data, please either upload them as Supporting Information files or deposit them to a stable, public repository and provide us with the relevant URLs, DOIs, or accession numbers. For a list of recommended repositories, please see https://journals.plos.org/plosone/s/recommended-repositories. If there are ethical or legal restrictions on sharing a de-identified data set, please explain them in detail (e.g., data contain potentially sensitive information, data are owned by a third-party organization, etc.) and who has imposed them (e.g., an ethics committee). Please also provide contact information for a data access committee, ethics committee, or other institutional body to which data requests may be sent. If data are owned by a third party, please indicate how others may request data access. 4. Please ensure that you include a title page within your main document. You should list all authors and all affiliations as per our author instructions and clearly indicate the corresponding author. 5. Please include your tables as part of your main manuscript and remove the individual files. Please note that supplementary tables (should remain/ be uploaded) as separate ""supporting information"" files 6. Please include captions for your Supporting Information files at the end of your manuscript, and update any in-text citations to match accordingly. Please see our Supporting Information guidelines for more information: http://journals.plos.org/plosone/s/supporting-information. 7. Please review your reference list to ensure that it is complete and correct. If you have cited papers that have been retracted, please include the rationale for doing so in the manuscript text, or remove these references and replace them with relevant current references. Any changes to the reference list should be mentioned in the rebuttal letter that accompanies your revised manuscript. If you need to cite a retracted article, indicate the article’s retracted status in the References list and also include a citation and full reference for the retraction notice.

Reviewers' comments:

Reviewer's Responses to Questions

**Comments to the Author**

1. Is the manuscript technically sound, and do the data support the conclusions?

Reviewer #1: Yes

Reviewer #2: Yes

2. Has the statistical analysis been performed appropriately and rigorously? 

Reviewer #1: N/A

Reviewer #2: N/A

3. Have the authors made all data underlying the findings in their manuscript fully available?

Reviewer #1: Yes

Reviewer #2: Yes

4. Is the manuscript presented in an intelligible fashion and written in standard English?

Reviewer #1: Yes

Reviewer #2: Yes

5. Review Comments to the Author

Reviewer #1: Dear authors

Thank you for your efforts to do this research. Below are some comments and suggestions to help improve the clarity and impact of your study.

Abstract:

The abstract is written in an unstructured format. If the journal guidelines permit, please revise it to a structured format with the following headings: Objectives, Methods, Results, Conclusion, Implications for practice, and Keywords.

Conclusions: Consider adding a sentence about the practical implications of your findings for healthcare providers.

Introduction section:

The introduction section is very lengthy. I strongly recommend it be written more concisely.

Additionally, the introduction should address the necessity of supporting parents by healthcare providers, including nurses. For example, the authors can refer to the following articles on this topic:

https://journals.sagepub.com/doi/full/10.1177/17455057221104674

https://onlinelibrary.wiley.com/doi/full/10.1155/2021/6697659

Discussion Section:

The discussion section provides a comprehensive overview of the study's aim, findings, and implications. However, some areas could benefit from further clarification, structural improvements, and more detailed integration of the study results with the existing literature.

Best Regards

Reviewer #2: The scoping review "Shared Book Reading in the NICU" is very interesting and covers all the specific issues important for the NICU.

The manuscript is well written and presented and could be accepted for publication in its current form.

6. PLOS authors have the option to publish the peer review history of their article (what does this mean? ). If published, this will include your full peer review and any attached files.

**Do you want your identity to be public for this peer review?** For information about this choice, including consent withdrawal, please see our Privacy Policy .

Reviewer #1: No

Reviewer #2: No

---

## [Author Response · Author response to Decision Letter 1]

4 Dec 2024

Sanjoy Kumer Dey, M.D

Academic Editor

PLOS ONE

Dear Dr. Dey:

Thank you for the review of our manuscript, “Promoting Language and Literacy Through Shared Book Reading in the NICU: A Scoping Review,” [PONE-D-23-44086]. We are pleased to learn your decision of “minor revision”, inviting us to re-submit our manuscript for publication.

We address your points and those of the reviewers below.

Journal Requirements

We have revised the manuscript to meet PLOS ONE’s style requirements.

2. Thank you for stating the following financial disclosure: "This research has been funded by a grant from the Deal Center for Early Language and Literacy to Lama K. Farran." Please state what role the funders took in the study. If the funders had no role, please state: ""The funders had no role in study design, data collection and analysis, decision to publish, or preparation of the manuscript."" If this statement is not correct you must amend it as needed. Please include this amended Role of Funder statement in your cover letter; we will change the online submission form on your behalf.

We have stated the following financial disclosure: “This research has been funded by a grant from the Deal Center for Early Language and Literacy to Lama K. Farran.”

We have stated the following role of the funders in the study: “The funders had no role in study design, data collection and analysis, decision to publish, or preparation of the manuscript.”

3. We note that your Data Availability Statement is currently as follows: All relevant data are within the manuscript and its Supporting Information files. Please confirm at this time whether or not your submission contains all raw data required to replicate the results of your study. Authors must share the “minimal data set” for their submission. PLOS defines the minimal data set to consist of the data required to replicate all study findings reported in the article, as well as related metadata and methods (https://journals.plos.org/plosone/s/data-availability#loc-minimal-data-set-definition).

We included our search strategy, including inclusionary and exclusionary criteria, as well as provided our PRISMA checklist. Additionally, we presented our findings, listing the number of articles meeting criteria for inclusion. Given the nature of this manuscript being a scoping review, the replicability of the findings is contingent upon using the same methodology at this date and time.

4. Please ensure that you include a title page within your main document. You should list all authors and all affiliations as per our author instructions and clearly indicate the corresponding author.

The title page is included listing all authors and their respective affiliations. We have specified the corresponding author.

5. Please include your tables as part of your main manuscript and remove the individual files. Please note that supplementary tables (should remain/ be uploaded) as separate ""supporting information"" files

We included the table as part of the main manuscript.

We included captions for our supporting information files at the end of our manuscript and updated the corresponding in-text citations.

We reviewed the reference list and none of the publications have been retracted. We have added new references as suggested by the reviewers.

We used PACE and verified that our figures meet PLOSONE requirements.

Response to Reviewer #1’s comments

1. The abstract is written in an unstructured format. If the journal guidelines permit, please revise it to a structured format with the following headings: Objectives, Methods, Results, Conclusion, Implications for practice, and Keywords.

We revised the abstract to include the suggested headings.

2. Conclusions: Consider adding a sentence about the practical implications of your findings for healthcare providers.

We added a sentence on the implications of our findings and the importance of including both caregivers and healthcare providers.

3. Introduction section: The introduction section is very lengthy. I strongly recommend it be written more concisely. Additionally, the introduction should address the necessity of supporting parents by healthcare providers, including nurses. For example, the authors can refer to the following articles on this topic:

https://journals.sagepub.com/doi/full/10.1177/17455057221104674

https://onlinelibrary.wiley.com/doi/full/10.1155/2021/6697659

We have shortened the introduction and addressed the necessity of healthcare providers’ (including nurses’) support for parents in the NICU. We cited the two articles the reviewers suggested.

4. The discussion section provides a comprehensive overview of the study's aim, findings, and implications. However, some areas could benefit from further clarification, structural improvements, and more detailed integration of the study results with the existing literature.

We have revised the discussion section to reflect a deeper integration of the study results with the existing literature.

Response to Reviewer #2’s comments

1. The manuscript is well written and presented and could be accepted for publication in its current form.

We appreciate your comments and positive feedback.

Sincerely,

Lama K. Farran

Susan Brasher

Sharon Leslie

---

## [Decision Letter · Decision Letter 1]

21 Jan 2025

Promoting Language and Literacy through Shared Book Reading in the NICU:

A Scoping Review

PONE-D-23-44086R1

Dear Dr. Lama K. Farran

We’re pleased to inform you that your manuscript has been judged scientifically suitable for publication and will be formally accepted for publication once it meets all outstanding technical requirements.

Kind regards,

Sanjoy Kumer Dey, M.D

Academic Editor

PLOS ONE
---

## [Editor Report · Acceptance letter]

PONE-D-23-44086R1

PLOS ONE

Dear Dr. Farran,

I'm pleased to inform you that your manuscript has been deemed suitable for publication in PLOS ONE. Congratulations! Your manuscript is now being handed over to our production team.

Kind regards,

on behalf of

Dr. Sanjoy Kumer Dey

Academic Editor

PLOS ONE